# ParMAC: Distributed Optimisation of Nested Functions, with Application to Learning Binary Autoencoders

**Miguel Á. Carreira-Perpiñán & Mehdi Alizadeh**
EECS, University of California, Merced
`http://eecs.ucmerced.edu`

## Abstract

Many powerful machine learning models are based on the composition of multiple processing layers, such as deep nets, which gives rise to nonconvex objective functions. A general, recent approach to optimise such "nested" functions is the *method of auxiliary coordinates (MAC)*. MAC introduces an auxiliary coordinate for each data point in order to decouple the nested model into independent submodels. This decomposes the optimisation into steps that alternate between training single layers and updating the coordinates. It has the advantage that it reuses existing single-layer algorithms, introduces parallelism, and does not need to use chain-rule gradients, so it works with nondifferentiable layers. We describe ParMAC, a distributed-computation model for MAC. This trains on a dataset distributed across machines while limiting the amount of communication so it does not obliterate the benefit of parallelism. ParMAC works on a cluster of machines with a circular topology and alternates two steps until convergence: one step trains the submodels in parallel using stochastic updates, and the other trains the coordinates in parallel. Only submodel parameters, no data or coordinates, are ever communicated between machines. ParMAC exhibits high parallelism, low communication overhead, and facilitates data shuffling, load balancing, fault tolerance and streaming data processing. We study the convergence of ParMAC and its parallel speedup, and implement ParMAC using MPI to learn binary autoencoders for fast image retrieval, achieving nearly perfect speedups in a 128-processor cluster with a training set of 100 million high-dimensional points.

## 1 Introduction

Serial computing has reached a plateau and parallel, distributed architectures are becoming widely available, from machines with a few cores to cloud computing with 1000s of machines. The combination of powerful nested models with large datasets is a key ingredient to solve difficult problems in machine learning, computer vision and other areas, and it underlies recent successes in deep learning (Hinton et al., 2012; Le et al., 2012; Dean et al., 2012). Unfortunately, parallel computation is not easy, and many good serial algorithms do not parallelise well. The cost of communicating (through the memory hierarchy or a network) greatly exceeds the cost of computing, both in time and energy, and will continue to do so for the foreseeable future. Thus, good parallel algorithms must minimise communication and maximise computation per machine, while creating sufficiently many subproblems (ideally independent) to benefit from as many machines as possible. The load (in runtime) on each machine should be approximately equal. Faults become more frequent as the number of machines increases, particularly if they are inexpensive machines. Machines may be heterogeneous and differ in CPU and memory; this is the case with initiatives such as SETI@home, which may become an important source of distributed computation in the future. Big data applications have additional restrictions. The size of the data means it cannot be stored on a single machine, so distributed-memory architectures are necessary. Sending data between machines is prohibitive because of the size of the data and the high communication costs. In some applications, more data is collected than can be stored, so data must be regularly discarded. In others, such as sensor networks, limited battery life and computational power imply that data must be processed locally.

In this paper, we focus on machine learning models of the form $\mathbf{y} = \mathbf{F}_{K+1}(\dots \mathbf{F}_2(\mathbf{F}_1(\mathbf{x}))\dots)$, i.e., consisting of a nested mapping from the input $\mathbf{x}$ to the output $\mathbf{y}$. Such *nested models* involve multiple parameterised layers of processing and include deep neural nets, cascades for object recognition in computer vision or for phoneme classification in speech processing, wrapper approaches to classification or regression, and various combinations of feature extraction/learning and preprocessing prior to some learning task. Nested and hierarchical models are ubiquitous in machine learning because they provide a way to construct complex models by the composition of simple layers. However, training nested models is difficult even in the serial case because *function composition produces inherently nonconvex functions*, which makes gradient-based optimisation difficult and slow, and sometimes inapplicable (e.g. with nonsmooth or discrete layers).

Our starting point is a recently proposed technique to train nested models, the *method of auxiliary coordinates (MAC)* (Carreira-Perpiñán and Wang, 2012; 2014). This reformulates the optimisation into an iterative procedure that alternates training submodels independently with coordinating them. It introduces significant model and data parallelism, can often train the submodels using existing algorithms, and has convergence guarantees with differentiable functions to a local stationary point, while it also applies with nondifferentiable or even discrete layers. MAC has been applied to various nested models (Carreira-Perpiñán and Wang, 2014; Wang and Carreira-Perpiñán, 2014; Carreira-Perpiñán and Raziperchikolaei, 2015; Raziperchikolaei and Carreira-Perpiñán, 2016; Carreira-Perpiñán and Vladymyrov, 2015). However, the original papers proposing MAC (Carreira-Perpiñán and Wang, 2012; 2014) did not address how to run MAC on a distributed computing architecture, where communication between machines is far costlier than computation. This paper proposes *ParMAC*, a parallel, distributed framework to learn nested models using MAC, analyses its parallel speedup and convergence, implements it in MPI for the problem of learning binary autoencoders, and demonstrates its ability to train on large datasets and achieve large speedups on a distributed cluster.

**Related work** Distributed optimisation and large-scale machine learning have been steadily gaining interest in recent years, with the development of parallel computation abstractions tailored to machine learning, such as Spark (Zaharia et al., 2010), GraphLab (Low et al., 2012), Petuum (Xing et al., 2015) or TensorFlow (Abadi et al., 2015), which have the goal of making cloud computing easily available to train machine learning models. Most work has centred on *convex* optimisation, particularly when the objective function has the form of empirical risk minimisation (data fitting term plus regulariser) (Cevher et al., 2014). This includes many important models in machine learning, such as linear regression, LASSO, logistic regression or SVMs. Such work is typically based on stochastic gradient descent (SGD) (Bottou, 2010), coordinate descent (CD) (Wright, 2016) or the alternating direction method of multipliers (ADMM) (Boyd et al., 2011). This has resulted in several variations of parallel SGD (Bertsekas, 2011; Zinkevich et al., 2010; Niu et al., 2011), parallel CD (Bradley et al., 2011; Richtárik and Takáč, 2013; Liu and Wright, 2015) and parallel ADMM (Boyd et al., 2011; Ouyang et al., 2013; Zhang and Kwok, 2014).

Little work has addressed *nonconvex* models. Most of it has focused on deep nets (Dean et al., 2012; Le et al., 2012). Google's DistBelief (Dean et al., 2012) uses asynchronous parallel SGD (with gradients for the full model computed with backpropagation) to achieve data parallelism, and some form of model parallelism. The latter is achieved by carefully partitioning the neural net into pieces and allocating them to machines to compute gradients. This is difficult to do and requires a careful match of the neural net structure (number of layers and hidden units, connectivity, etc.) to the target hardware. Also, parallel SGD can diverge with nonconvex models, which requires heuristics to make sure we average replica models that are close in parameter space and thus associated with the same optimum. Although this has managed to train huge nets on huge datasets by using tens of thousands of CPU cores, the speedups achieved were very modest. Other work has used similar techniques but for GPUs (Coates et al., 2013; Seide et al., 2014).

Finally, there also exist specific approximation techniques for certain types of large-scale machine learning problems, such as spectral problems, using the Nyström formula or other landmark-based methods (Williams and Seeger, 2001; Bengio et al., 2004; Drineas and Mahoney, 2005; Talwalkar et al., 2008; Vladymyrov and Carreira-Perpiñán, 2013; 2016).

*ParMAC is specifically designed for nested models, which are typically nonconvex and include deep nets and many other models, some of which have nondifferentiable layers.* As we describe below, ParMAC has the advantages of being simple and relatively independent of the target hardware, while achieving high speedups.

## 2   Optimising nested models using auxiliary coordinates (MAC)

Many optimisation problems in machine learning involve mathematically "nested" functions of the form $\mathbf{F}(\mathbf{x}; \mathbf{W}) = \mathbf{F}_{K+1}(\ldots \mathbf{F}_2(\mathbf{F}_1(\mathbf{x}; \mathbf{W}_1); \mathbf{W}_2) \ldots; \mathbf{W}_{K+1})$ with parameters $\mathbf{W}$, such as deep nets. Such problems are traditionally optimised using methods based on gradients computed using the chain rule. However, such gradients may sometimes be inconvenient to use, or may not exist (e.g. if some of the layers are nondifferentiable, as with binary autoencoders). Also, they are hard to parallelise, because of the inherent sequentiality in the chain rule. The *method of auxiliary coordinates (MAC)* (Carreira-Perpiñán and Wang, 2012; 2014) is designed to optimise nested models without using chain-rule gradients while introducing parallelism. The idea is to break nested functional relationships judiciously by introducing new variables (the *auxiliary coordinates*) as equality constraints. These are then solved by optimising a penalised function using alternating optimisation over the original parameters (which we call the $\mathbf{W}$ step) and over the coordinates (which we call the $\mathbf{Z}$ step). The result is a *coordination-minimisation (CM) algorithm*: the minimisation ($\mathbf{W}$) step updates the parameters by splitting the nested model into independent submodels and training them using existing algorithms, and the coordination ($\mathbf{Z}$) step ensures that corresponding inputs and outputs of submodels eventually match. MAC algorithms have been developed for several nested models so far: deep nets (Carreira-Perpiñán and Wang, 2014), low-dimensional SVMs (Wang and Carreira-Perpiñán, 2014), binary autoencoders (Carreira-Perpiñán and Raziperchikolaei, 2015), affinity-based loss functions for binary hashing (Raziperchikolaei and Carreira-Perpiñán, 2016) and parametric nonlinear embeddings (Carreira-Perpiñán and Vladymyrov, 2015). Although this paper proposes and analyses ParMAC in general, our MPI implementation is for the particular case of binary autoencoders. These define a nonconvex nondifferentiable problem, yet its MAC algorithm is simple and effective.

**MAC algorithm for binary autoencoders**   A *binary autoencoder (BA)* is a usual autoencoder but with a binary code layer. It consists of an encoder $\mathbf{h}(\mathbf{x})$ that maps a real vector $\mathbf{x} \in \mathbb{R}^D$ onto a *binary* code vector with $L < D$ bits, $\mathbf{z} \in \{0,1\}^L$, and a linear decoder $\mathbf{f}(\mathbf{z})$ which maps $\mathbf{z}$ back to $\mathbb{R}^D$ in an effort to reconstruct $\mathbf{x}$. We will call $\mathbf{h}$ a *binary hash function* (see later). Let us write $\mathbf{h}(\mathbf{x}) = \lceil(\mathbf{A}\mathbf{x})$ ($\mathbf{A}$ includes a bias by having an extra dimension $x_0 = 1$ for each $\mathbf{x}$) where $\mathbf{A} \in \mathbb{R}^{L \times (D+1)}$ and $\lceil(t)$ is a step function applied elementwise, i.e., $\lceil(t) = 1$ if $t \geq 0$ and $\lceil(t) = 0$ otherwise. Given a dataset of $D$-dimensional patterns $\mathbf{X} = (\mathbf{x}_1, \ldots, \mathbf{x}_N)$, our objective function, which involves the nested model $\mathbf{y} = \mathbf{f}(\mathbf{h}(\mathbf{x}))$, is the usual least-squares reconstruction error $E_{\mathrm{BA}}(\mathbf{h}, \mathbf{f}) = \sum_{n=1}^N \|\mathbf{x}_n - \mathbf{f}(\mathbf{h}(\mathbf{x}_n))\|^2$. Optimising this nonconvex, nonsmooth function is NP-complete. Where the gradients do exist wrt $\mathbf{A}$ they are zero, so optimisation of $\mathbf{h}$ using chain-rule gradients does not apply. We introduce as auxiliary coordinates the outputs of $\mathbf{h}$, i.e., the codes for each of the $N$ input patterns, and obtain the following equality-constrained problem:

$$\min_{\mathbf{h},\mathbf{f},\mathbf{Z}} \sum_{n=1}^N \|\mathbf{x}_n - \mathbf{f}(\mathbf{z}_n)\|^2 \quad \text{s.t.} \quad \mathbf{z}_n = \mathbf{h}(\mathbf{x}_n), \ \mathbf{z}_n \in \{0,1\}^L, \ n = 1, \ldots, N. \tag{1}$$

Note the codes are binary. We now apply the quadratic-penalty method and minimise the following objective function while progressively increasing $\mu$, so the constraints are eventually satisfied:

$$E_Q(\mathbf{h}, \mathbf{f}, \mathbf{Z}; \mu) = \sum_{n=1}^N \|\mathbf{x}_n - \mathbf{f}(\mathbf{z}_n)\|^2 + \mu \|\mathbf{z}_n - \mathbf{h}(\mathbf{x}_n)\|^2 \ \text{s.t.} \ \mathbf{z}_n \in \{0,1\}^L, \ n = 1, \ldots, N. \tag{2}$$

Finally, we apply alternating optimisation over $\mathbf{Z}$ and $\mathbf{W} = (\mathbf{h}, \mathbf{f})$. This gives the following steps:

- Over $\mathbf{Z}$ for fixed $(\mathbf{h}, \mathbf{f})$, this is a binary optimisation on $NL$ variables, but it separates into $N$ independent optimisations each on only $L$ variables, with the form of a binary proximal operator (where we omit the index $n$): $\min_{\mathbf{z}} \|\mathbf{x} - \mathbf{f}(\mathbf{z})\|^2 + \mu \|\mathbf{z} - \mathbf{h}(\mathbf{x})\|^2$ s.t. $\mathbf{z} \in \{0,1\}^L$. This can be solved approximately by alternating optimisation over bits.
- Over $\mathbf{W} = (\mathbf{h}, \mathbf{f})$ for fixed $\mathbf{Z}$, we obtain $L + D$ independent problems: for each of the $L$ single-bit hash functions (which try to predict $\mathbf{Z}$ optimally from $\mathbf{X}$), each solvable by fitting a linear SVM; and for each of the $D$ linear decoders in $\mathbf{f}$ (which try to reconstruct $\mathbf{X}$ optimally from $\mathbf{Z}$), each a linear least-squares problem.

The user must choose a schedule for the penalty parameter $\mu$ (sequence of values $0 < \mu_1 < \cdots < \mu_\infty$). This should increase slowly enough that the binary codes can change considerably and explore

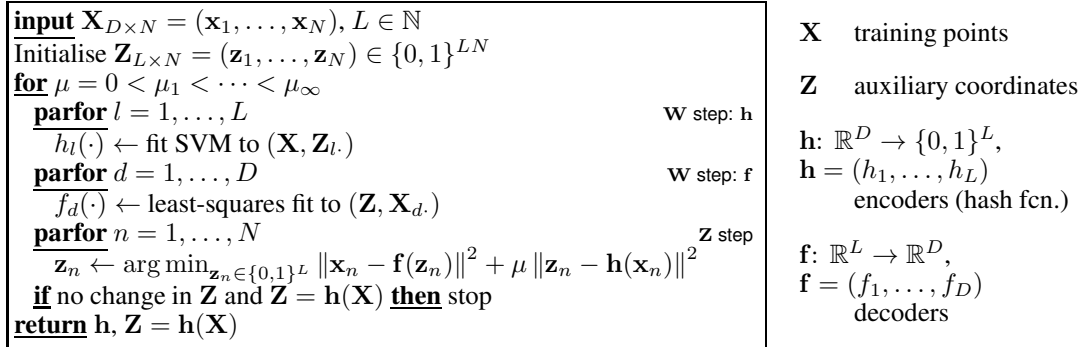

Figure 1: MAC algorithm for binary autoencoders. "**parfor**" indicates a **for** loop whose iterations are carried out in parallel. The steps over $\mathbf{h}$ and $\mathbf{f}$ can be run in parallel as well.

better solutions before the constraints are satisfied and the algorithm stops. With BAs, MAC stops for a finite value of $\mu$, which occurs whenever $\mathbf{Z}$ does not change compared to the previous $\mathbf{Z}$ step. This gives a practical stopping criterion. Carreira-Perpiñán and Raziperchikolaei (2015) give proofs of these statements and further details about the algorithm. Fig. 1 gives the MAC algorithm for BAs.

The BA was proposed as a way to learn good binary hash functions for fast, approximate information retrieval (Carreira-Perpiñán and Raziperchikolaei, 2015). Binary hashing (Grauman and Fergus, 2013) has emerged in recent years as an effective way to do fast, approximate nearest-neighbour searches in image databases. The real-valued, high-dimensional image vectors are mapped onto a binary space with $L$ bits and the search is performed there using Hamming distances at a vastly faster speed and smaller memory (e.g. $N = 10^9$ points with $D = 500$ take 2 TB, but only 8 GB using $L = 64$ bits, which easily fits in RAM). As shown by Carreira-Perpiñán and Raziperchikolaei (2015), training BAs with MAC beats approximate optimisation approaches such as relaxing the codes or the step function in the encoder, and yields state-of-the-art binary hash functions $\mathbf{h}$ in unsupervised problems, improving over established approaches such as iterative quantisation (ITQ) (Gong et al., 2013). We focus mostly on linear hash functions because these are, by far, the most used type of hash functions in the literature of binary hashing, due to the fact that computing the binary codes for a test image must be fast at run time.

**MAC in general** With a nested function with $K$ layers, we can introduce auxiliary coordinates at each layer. For example, with a neural net, this decouples the weight vector of every hidden unit in the $\mathbf{W}$ step, which can be solved as a logistic regression (see Carreira-Perpiñán and Alizadeh, 2016). For a large net with a large dataset, this affords an enormous potential for parallel computation.

**MAC and EM** MAC is very similar to expectation-maximisation (EM) at a conceptual level. EM (McLachlan and Krishnan, 2008) applies generally to many probabilistic models. The resulting algorithm can be very different (e.g. EM for Gaussian mixtures vs EM for hidden Markov models), but it always alternates two steps that conceptually do the following. The E step updates in parallel the posterior probabilities. This separates over data points and is like the $\mathbf{Z}$ step in MAC, where the posterior probabilities are the auxiliary coordinates, and where the step may be in closed-form or require optimisation, depending on the model. The M step updates in parallel the "submodels". For a mixture with $M$ components, these are the $M$ Gaussians (means, covariances, proportions). This separates over submodels and is like the $\mathbf{W}$ step in MAC. For BAs, the submodels are the $L$ encoders (linear SVMs) and the $D$ decoders (linear regressors); for a neural net, each weight vector of a hidden unit is a submodel (a logistic regressor). For Gaussian mixtures, the M step can be done exactly in one "epoch" because it is a simple average. For MAC, it usually requires optimisation, and so multiple epochs. In fact, ParMAC applies to EM by using $e = 1$ epoch: in the $\mathbf{W}$ step, the Gaussians visit each machine circularly and (their averages) are updated on its data; in the $\mathbf{Z}$ step, each machine updates its posterior probabilities.

In the rest of the paper, some readers may find this analogy useful and think of EM for Gaussian mixtures instead of MAC, replacing "submodels" and "auxiliary coordinates" in MAC with "Gaussians" and "posterior probabilities" in EM, respectively.

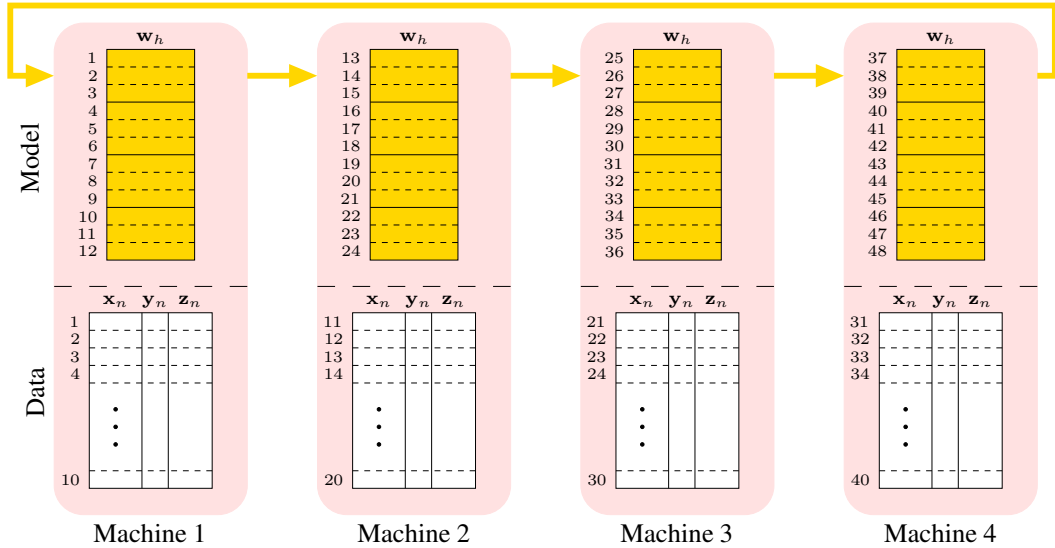

Figure 2: ParMAC model with $P = 4$ machines, $M = 12$ submodels "$\mathbf{w}_h$" and $N = 40$ data points. Submodels $h$, $h + M$, $h + 2M$ and $h + 3M$ are copies of submodel $h$, but only one of them is the most currently updated. At the end of the $\mathbf{W}$ step all copies are identical.

## 3 PARMAC: A PARALLEL, DISTRIBUTED COMPUTATION MODEL FOR MAC

A specific MAC algorithm depends on the model and objective function and on how the auxiliary coordinates are introduced. We can achieve steps that are closed-form, convex, nonconvex, binary, or others. However, we will assume the following always hold: (1) **Separability over data points**. In the $\mathbf{Z}$ step, *the $N$ subproblems for $\mathbf{z}_1, \dots, \mathbf{z}_N$ are independent, one per data point*. Each $\mathbf{z}_n$ step depends on the current model. (2) **Separability over submodels**. In the $\mathbf{W}$ step, there are *$M$ independent submodels*, where $M$ depends on the problem. For example, $M$ is the number of hidden units in a deep net, or the number of hash functions and linear decoders in a BA. Each submodel depends on all the data and coordinates. We now show how to turn this into a distributed, low-communication *ParMAC* algorithm.

The basic idea in ParMAC is as follows. With large datasets in distributed systems, it is imperative to minimise data movement over the network because the communication time generally far exceeds the computation time in modern architectures. In MAC we have 3 types of data: the original training data $(\mathbf{X}, \mathbf{Y})$, the auxiliary coordinates $\mathbf{Z}$, and the model parameters (the submodels). Usually, the latter type is far smaller. *In ParMAC, we never communicate training or coordinate data; each machine keeps a disjoint portion of $(\mathbf{X}, \mathbf{Y}, \mathbf{Z})$ corresponding to a subset of the points. Only model parameters are communicated, during the $\mathbf{W}$ step, following a circular topology, which implicitly implements a stochastic optimisation*. The model parameters are the hash functions $\mathbf{h}$ and the decoder $\mathbf{f}$ for BAs, and the weight vector $\mathbf{w}_h$ of each hidden unit $h$ for deep nets. Let us see this in detail (refer to fig. 2).

Assume we have $P$ identical processing machines, each with its own memory and CPU, connected through a network in a circular unidirectional topology. Each machine stores a subset of the data points and corresponding coordinates $(\mathbf{x}_n, \mathbf{y}_n, \mathbf{z}_n)$ such that the subsets are disjoint and their union is the entire data. Before the $\mathbf{Z}$ step starts, each machine contains all the (just updated) submodels. This means that in the $\mathbf{Z}$ step each machine processes its auxiliary coordinates $\{\mathbf{z}_n\}$ independently of all other machines, i.e., no communication occurs. The $\mathbf{W}$ step is more subtle. At the beginning of the $\mathbf{W}$ step, each machine will contain all the submodels and its portion of the data and (just updated) coordinates. Each submodel must have access to the entire data and coordinates in order to update itself and, since the data cannot leave its home machine, the submodel must go to the data. We achieve this in the circular topology with an asynchronous processing, as follows. Each machine keeps a queue of submodels to be processed, and repeatedly performs the following operations: extract a submodel from the queue, process it on its data and send it to the machine's successor

(which will insert it in its queue). If the queue is empty, the machine waits until it is nonempty. The queue of each machine is initialised with a portion $M/P$ of submodels associated with that machine (e.g. in fig. 2, machine 1's queue contains submodels 1–3, machine 2 submodels 4–6, etc.). Each submodel carries a counter that is initially 1 and increases every time it visits a machine. When it reaches $P$, the submodel has visited all machines in sequence and has completed an *epoch*. We repeat this for $e$ epochs and, to ensure all machines have all final submodels before starting the **Z** step, we run a communication-only epoch $e + 1$ (without computation), where submodels simply move from machine to machine.

Since each submodel is updated as soon as it visits a machine, rather than computing the exact gradient once it has visited all machines and then take a step, the **W** step is really carrying out *stochastic steps for each submodel*. For example, if the update is done by a gradient step, we are actually implementing stochastic gradient descent (SGD) where the minibatches are of size $N/P$ (or smaller, if we subdivide a machine's data portion into minibatches, which should be typically the case in practice). From this point of view, we can regard the **W** step as doing SGD on each submodel in parallel by having each submodel visit the minibatches in each machine.

As described, and as implemented in our experiments, the entire model parameters are communicated $e + 1$ times in a MAC iteration if running $e$ epochs in the **W** step. We can also run $e$ epochs with only 2 rounds of communication *by having a submodel do $e$ consecutive passes within each machine's data*. This reduces the amount of shuffling, but should not be a problem if the data are randomly distributed over machines.

**Extensions of ParMAC**  *Data shuffling*, which improves the SGD convergence speed, can be achieved without data movement by accessing the local data in random order at each epoch (within-machine), and by randomising the circular topology at each epoch (across-machine). *Load balancing* is simple because the work in both **W** and **Z** steps is proportional to the number of data points $N$. Hence, if the processing power of machine $p$ is proportional to $\alpha_p > 0$, we allocate to it $N\alpha_p/(\alpha_1 + \cdots + \alpha_P)$ data points. *Streaming*, i.e., discarding old data and adding new data during training, can be done by adding/removing data within-machine, or by adding/removing machines and updating the circular topology. *Fault tolerance* is possible because we can still learn a good model even if we lose the data from a machine that fails, and because in the **W** step we can revert to older copies of the lost submodels residing in other machines. See further details in Carreira-Perpiñán and Alizadeh (2016).

**A theoretical model of the parallel speedup**  We can estimate the runtime of the **W** and **Z** steps assuming there are $M$ independent submodels of the same size in the **W** step, using $e$ epochs, on a dataset with $N$ training points, distributed over $P$ identical machines (each with $N/P$ points). Let $t_r^{\mathbf{W}}$ be the computation time per submodel and data point in the **W** step, $t_r^{\mathbf{Z}}$ the computation time per data point in the **Z** step, and $t_c^{\mathbf{W}}$ the communication time per submodel in the **W** step. Then the runtime of the **W** and **Z** steps is $T^{\mathbf{W}}(P) = \lceil M/P\rceil(t_r^{\mathbf{W}}\frac{N}{P} + t_c^{\mathbf{W}})Pe + \lceil M/P\rceil t_c^{\mathbf{W}}P$ and $T^{\mathbf{Z}}(P) = M\frac{N}{P}t_r^{\mathbf{Z}}$, respectively. Hence the parallel speedup is (see details in Carreira-Perpiñán and Alizadeh, 2016):

$$S(P) = \frac{T(1)}{T(P)} = \frac{\rho\frac{1}{\lceil M/P\rceil}MP}{\frac{1}{N}P^2 + \rho_2 P + \rho_1\frac{1}{\lceil M/P\rceil}M} \qquad \begin{array}{l} \rho_1 = t_r^{\mathbf{Z}}/(e+1)t_c^{\mathbf{W}}, \ \rho_2 = et_r^{\mathbf{W}}/(e+1)t_c^{\mathbf{W}} \\ \rho = \rho_1 + \rho_2 = (et_r^{\mathbf{W}} + t_r^{\mathbf{Z}})/(e+1)t_c^{\mathbf{W}} \end{array} \quad (3)$$

where $\rho$, $\rho_1$ and $\rho_2$ are ratios of computation vs communication, dependent on the optimisation algorithm in the **W** and **Z** steps, and on the performace of the distributed system and MPI library.

Hence, if $P \leq M$ and $M$ is divisible by $P$ we have $S(P) = P/(1 + \frac{P}{\rho N})$ and if $P > M$ we have $S(P) = \rho M/(\rho_2 + \rho_1\frac{M}{P} + \frac{P}{N})$. In practice, typically we have $\rho \ll 1$ (because communication dominates computation in current architectures) and $\rho_2 N \gg 1$ (large dataset). If we take $P \ll \rho_2 N$, then $S(P) \approx P$ if $P \leq M$ and $S(P) \approx \rho M/(\rho_2 + \rho_1\frac{M}{P})$ if $P > M$. Hence, *the speedup is nearly perfect if using fewer machines than submodels, and otherwise it peaks at $S_1^* = \rho M/(\rho_2 + 2\sqrt{\rho_1 M/N}) > M$ for $P = P_1^* = \sqrt{\rho_1 MN} > M$ and decreases thereafter*. This affords very large speedups for large datasets and large models. This theoretical speedup matches well our measured ones (see the experiments section), and can be used to determine optimal values for the number of machines $P$ to use in practice (subject to additional constraints, e.g. cost of the machines).

Eq. (3) also shows that we can leave the speedup unchanged by trading off dataset size and computation/communication times, as long as one of these holds: $Nt_r^{\mathbf{W}}$ and $Nt_r^{\mathbf{Z}}$ remain constant; or $N/t_c^{\mathbf{W}}$ remains constant; or $t_r^{\mathbf{W}}/t_c^{\mathbf{W}}$ and $t_r^{\mathbf{Z}}/t_c^{\mathbf{W}}$ remain constant.

In the BA, we have submodels of different size: encoders of size $D$ and decoders of size $L < D$. We can model this by "grouping" the $D$ decoders into $L$ groups of $D/L$ decoders each, resulting in $M = 2L$ equal-size submodels (assuming the ratio of computation and communication times of decoder vs encoder is $L/D < 1$).

**Convergence of ParMAC** The only approximation that ParMAC makes to the original MAC algorithm is using SGD in the $\mathbf{W}$ step. Since we can guarantee convergence of SGD under certain conditions (e.g. Robbins-Monro schedules), we can recover the original convergence guarantees for MAC to a local stationary point with differentiable layers (see details in Carreira-Perpiñán and Alizadeh, 2016). This convergence guarantee is independent of the number of layers, models and processors. With nondifferentiable layers, the convergence properties of MAC (and ParMAC) are not well known. In particular, for the binary autoencoder the encoding layer is discrete and the problem is NP-complete. While convergence guarantees are important theoretically, in practical applications with large datasets in a distributed setting one typically runs SGD for just a few epochs, even one or less than one (i.e., we stop SGD before passing through all the data). This typically reduces the objective function to a good enough value as fast as possible, since each pass over the data is very costly. In our experiments, 1–2 epochs in the $\mathbf{W}$ step make ParMAC very similar to MAC using an exact step.

**Circular vs parameter-server topologies** We also considered implementing ParMAC using a parameter-server (PS) topology rather than a circular one, but the latter is better. With a PS we do parallel SGD on each submodel independently, i.e., each worker runs SGD on its own submodel replica for a while, sends it to the PS, and this broadcasts an "average" submodel back to the workers, asynchronously. The circular topology does true SGD on each submodel independently from the others. We can show the runtime per iteration using a PS is equal to that of the circular topology only if the server can communicate with $P$ workers simultaneously (rather than sequentially), otherwise it is slower. The reason is the PS has more communication. The PS has some additional disadvantages: parallel SGD converges more slowly than true SGD and is difficult to apply if the $\mathbf{W}$ step is nonconvex; and it needs extra machine(s) to act as parameter server(s). The fundamental issue is that both topologies differ in how they employ the available parallelism: the circular topology updates different, independent submodels, while the PS updates replicas of the same submodels.

## 4 EXPERIMENTS

*MPI implementation of ParMAC for BAs.* We have used C/C++, the GSL and BLAS libraries for mathematical operations, and the Message Passing Interface (MPI) (Gropp et al., 1999) for interprocess communication. MPI is a widely used framework for high-performance parallel computing, available in multiple platforms. It is particularly suitable for ParMAC because of its support of the SPMD (single program, multiple data) model. In MPI, processes in different machines communicate through messages. To receive data, we use the synchronous blocking receive function `MPI_Recv`; the process calling this blocks until the data arrives. To send data we use the buffered blocking send function `MPI_Bsend`. We allocate enough memory and attach it to the system. The process calling `MPI_Bsend` blocks until the buffer is copied to the MPI internal memory; after that, the MPI library takes care of sending the data. See a code snippet in Carreira-Perpiñán and Alizadeh (2016).

*Distributed-memory cluster.* We used General Computing Nodes from the UCSD Triton Shared Computing Cluster (TSCC), available to the public for a fee. Each node contains 2 8-core Intel Xeon E5-2670 processors (16 cores in total), 64GB RAM (4GB/processor) and a 500GB hard drive. The nodes are connected through a 10GbE network. We used up to $P = 128$ processors. Carreira-Perpiñán and Alizadeh (2016) give detailed specs as well as experiments in a shared-memory machine.

*Datasets.* We have used 3 well-known colour image retrieval benchmarks. (1) CIFAR (Krizhevsky, 2009) contains 60 000 images ($N = 50\,000$ training and 10 000 test), represented by $D = 320$ GIST features. (2) SIFT-1M (Jégou et al., 2011a) contains $N = 10^6$ training and $10^4$ test images, each

represented by $D = 128$ SIFT features. (3) SIFT-1B (Jégou et al., 2011a) has three subsets: $10^9$ base vectors where the search is performed, $N = 10^8$ learning vectors used to train the model and $10^4$ query vectors.

*Performance measures.* Regarding the quality of the BA and hash functions learnt, we report the retrieval precision (%) in the test set using as true neighbours the $K$ nearest images in Euclidean distance in the original space, and as retrieved neighbours in the binary space we use the $k$ nearest images in Hamming distance. We set $(K, k) = (1\,000, 100)$ for CIFAR and $(10\,000, 10\,000)$ for SIFT-1M. For SIFT-1B, as suggested by the dataset creators, we report the recall@R: the average number of queries for which the nearest neighbour is ranked within the top $R$ positions (for varying values of $R$); in case of tied distances, we place the query as top rank. All these measures are computed offline once the BA is trained. Carreira-Perpiñán and Alizadeh (2016) give additional measures and experiments.

*Models and their parameters.* We use BAs with linear encoders (linear SVM) except with SIFT-1B, where we also use kernel SVMs. The decoder is always linear. We set $L = 16$ bits (hash functions) for CIFAR and SIFT-1M and $L = 64$ bits for SIFT-1B. We initialise the binary codes from truncated PCA ran on a subset of the training set (small enough that it fits in one processor). To train the encoder ($L$ SVMs) and decoder ($D$ linear mappings) with stochastic optimisation, we used the SGD code from (Bottou and Bousquet, 2008), using its default parameter settings. The SGD step size is tuned automatically in each iteration by examining the first $1\,000$ datapoints. We use a multiplicative $\mu$ schedule $\mu_i = \mu_0 a^i$ where the initial value $\mu_0$ and the factor $a > 1$ are tuned offline in a trial run using a small subset of the data. For CIFAR we use $\mu_0 = 0.005$ and $a = 1.2$ over 26 iterations ($i = 0, \ldots, 25$). For SIFT-1M and SIFT-1B we use $\mu_0 = 10^{-4}$ and $a = 2$ over 10 iterations.

**Effect of stochastic steps in the W step** Fig. 3 shows the effect on the precision on CIFAR of varying the number of epochs within the **W** step and shuffling the data as a function of the number of processors $P$. As the number of epochs increases, the **W** step is solved more exactly (8 epochs is practically exact in this data). Fewer epochs, even just one, cause only a small degradation. The reason is that, although these are relatively small datasets, they contain sufficient redundancy that few epochs are sufficient to decrease the error considerably.

This is also helped by the accumulated effect of epochs over MAC iterations. Running more epochs increases the runtime and lowers the parallel speedup in this particular model, because we use few bits ($L = 16$) and therefore few submodels ($M = 2L = 32$) compared to the number of machines (up to $P = 128$), so the **W** step has less parallelism. The positive effect of data shuffling in the **W** step is clear: shuffling generally increases the precision with no increase in runtime.

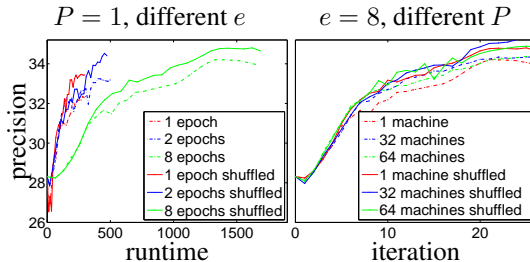

Figure 3: Precision in CIFAR dataset.

**Speedup** The fundamental advantage of ParMAC and distributed optimisation in general is the ability to train on datasets that do not fit in a single machine, and the reduction in runtime because of parallel processing. Fig. 4 shows the "strong scaling" speedups achieved, as a function of the number of machines $P$ for fixed problem size (dataset and model), in CIFAR and SIFT-1M ($N = 50$K and 1M training points, respectively). Even though these datasets and especially the number of independent submodels ($M = 2L = 32$ effective submodels of the same size, as discussed earlier) are relatively small, the speedups we achieve are nearly perfect for $P \leq M$ and hold very well for $P > M$ up to the maximum number of machines we used ($P = 128$ in the distributed system). The speedups flatten as the number of **W**-step epochs (and consequently the amount of communication) increases, because for this experiment the bottleneck is the **W** step, whose parallelisation ability (i.e., the number of concurrent processes) is limited by $M = 2L$ (the **Z** step has $N$ independent processes and is never a bottleneck, since $N$ is very large). However, as noted earlier, using 1 to 2 epochs gives a good enough result, very close to doing an exact **W** step. The runtime for SIFT-1M on $P = 128$ machines with 8 epochs was 12 minutes and its speedup $100\times$. This is particularly remarkable given that the original, nested model did not have model parallelism.

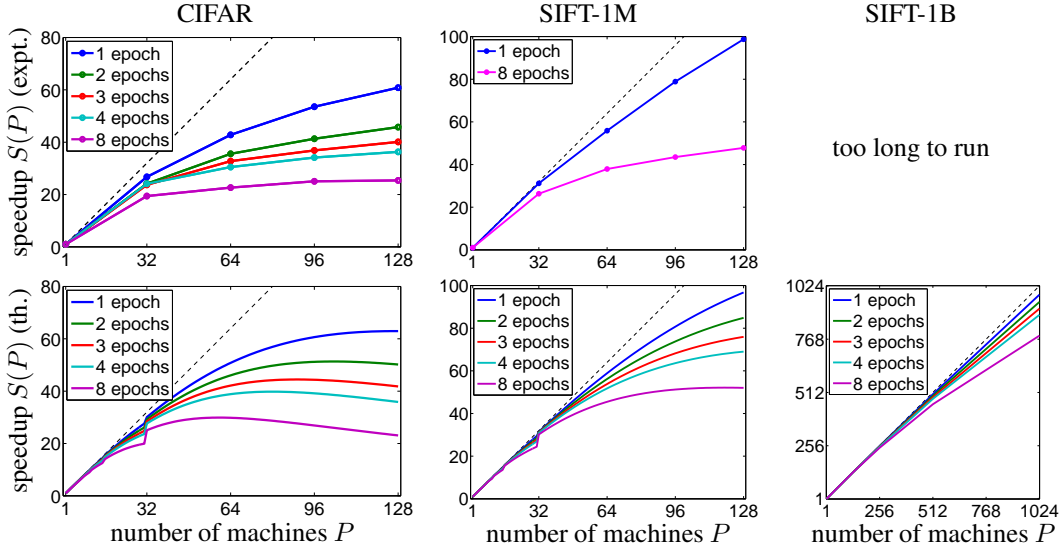

Figure 4: Speedup $S(P)$ as a function of the number of machines $P$ (top: experiment, bottom: theory). The dataset size and number of submodels $(N, M)$ is $(50\,000, 32)$ for CIFAR, $(10^6, 32)$ for SIFT-1M and $(10^8, 128)$ for SIFT-1B.

Fig. 4 also shows the speedups predicted by our theoretical model. We set the parameters $e$ and $N$ to their known values, and $M = 2L = 32$ for CIFAR and SIFT-1M and $M = 2L = 128$ for SIFT-1B. For the time parameters, we set $t_r^{\mathbf{W}} = 1$ to fix the time units, and we set $t_c^{\mathbf{W}}$ and $t_r^{\mathbf{Z}}$ by trial and error to achieve a reasonably good fit to the experimental speedups: $t_c^{\mathbf{W}} = 10^4$ for both datasets, and $t_r^{\mathbf{Z}} = 200$ for CIFAR and $40$ for SIFT-1M. Although these are fudge factors, they are in rough agreement with the fact that communicating a weight vector over the network is orders of magnitude slower than updating it with a gradient step, and that the $\mathbf{Z}$ step is quite slower than the $\mathbf{W}$ step because of the binary optimisation it involves.

**Large-scale experiment**  SIFT-1B is one of the largest datasets, if not the largest one, that are publicly available for comparing nearest-neighbour search algorithms with known ground-truth (i.e., precomputed exact Euclidean distances for each query to its $k$ nearest vectors in the base set). The training set contains $N = 100$M vectors, each consisting of 128 SIFT features. We used $L = 64$ hash functions ($M = 128$ submodels): linear SVMs as before, and kernel SVMs. These have fixed Gaussian radial basis functions (2 000 centres picked at random from the training set and bandwidth $\sigma = 160$), so the only trainable parameters are the weights, and the MAC algorithm does not change except that it operates on a 2 000-dimensional input vector of kernel values, instead of the 128 SIFT features. We use $e = 2$ epochs with shuffling. All these decisions were based on trials on a subset of the training dataset. We initialised the binary codes from truncated PCA trained on a subset of size 1M (recall@R=100: 55.2%), which gave results comparable to the baseline in (Jégou et al., 2011b).

We ran ParMAC on the whole training set in the distributed system with 128 processors for 6 iterations and achieved a recall@R=100 of 61.5% in 29 hours (linear SVM) and 66.1% in 83 hours (kernel SVM). Using a scaled-down model and training set, we estimated that training in one machine (with enough RAM to hold the data and parameters) would take months. The theoretical speedup (fig. 4 right plot, using the same parameters as in SIFT-1M), is nearly perfect (note the plot goes up to $P = 1\,024$ machines, even though our experiments are limited to $P = 128$). This is because $M$ is quite larger and $N$ is much larger than in the previous datasets.

## 5 DISCUSSION

Developing parallel, distributed optimisation algorithms for nonconvex problems in machine learning is challenging, as shown by recent efforts by large teams of researchers (Le et al., 2012; Dean et al., 2012). One important advantage of ParMAC is its simplicity. Data and model paral-

lelism arise naturally thanks to the introduction of auxiliary coordinates. The corresponding optimisation subproblems can often be solved reusing existing code as a black box (as with the SGD training of SVMs and linear mappings in the BA). A circular topology is sufficient to achieve a low communication between machines. There is no close coupling between the model structure and the distributed system architecture. This makes ParMAC suitable for architectures as different as supercomputers and data centres.

Further improvements can be made in specific problems. For example, we may have more parallelisation or less dependencies (e.g. the weights of hidden units in layer $k$ of a neural net depend only on auxiliary coordinates in layers $k$ and $k + 1$). This may reduce the communication in the $\mathbf{W}$ step, by sending to a given machine only the model portion it needs, or by allocating cores within a multicore machine accordingly. The $\mathbf{W}$ and $\mathbf{Z}$ step optimisations can make use of further parallelisation by GPUs or by distributed convex optimisation algorithms. Many more refinements can be done, such as storing or communicating reduced-precision values with little effect of the accuracy. In this paper, we have tried to keep our implementation as simple as possible, because our goal was to understand the parallelisation speedups of ParMAC in a setting as general as possible, rather than trying to achieve the very best performance for a particular dataset, model or distributed system.

## 6 CONCLUSION

We have proposed ParMAC, a distributed model for the method of auxiliary coordinates for training nested, nonconvex models in general, analysed its parallel speedup and convergence, and demonstrated it with an MPI-based implementation for a particular case, to train binary autoencoders. MAC creates parallelism by introducing auxiliary coordinates for each data point to decouple nested terms in the objective function. ParMAC is able to translate the parallelism inherent in MAC into a distributed system by 1) using data parallelism, so that each machine keeps a portion of the original data and its corresponding auxiliary coordinates; and 2) using model parallelism, so that independent submodels visit every machine in a circular topology, effectively executing epochs of a stochastic optimisation, without the need for a parameter server and therefore no communication bottlenecks. The convergence properties of MAC (to a stationary point of the objective function) remain essentially unaltered in ParMAC. The parallel speedup can be theoretically predicted to be nearly perfect when the number of submodels is comparable or larger than the number of machines, and to eventually saturate as one continues to increase the number of machines, and indeed this was confirmed in our experiments. ParMAC also makes it easy to account for data shuffling, load balancing, streaming and fault tolerance. Hence, we expect that ParMAC could be a basic building block, in combination with other techniques, for the distributed optimisation of nested models in big data settings.

### ACKNOWLEDGMENTS

Work supported by a Google Faculty Research Award and by NSF award IIS–1423515. We thank Ramin Raziperchikolaei (UC Merced) for discussions about binary autoencoders, Dong Li (UC Merced) for discussions about MPI and performance evaluation on parallel systems, and Quoc Le (Google) for discussions about Google's DistBelief system.

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
