# Peer review of "ParMAC: distributed optimisation of nested functions, with application to binary autoencoders"

_ICLR 2017 — rejected_

[Official Review · AnonReviewer2 · rating 6 · confidence 4 · 19 Dec 2016]
**good paper but need some clarification on MAC**

This paper proposes a novel approach ParMAC, a parallel and distributed framework of MAC (the Method of Auxiliary Coordinates) to learn nested and non-convex models which is based on the composition of multiple processing layers (i.e., deep nets). The basic idea of MAC to optimise the nested objective function, which is traditionally learned using methods based on the chain-rule gradients but inconvenient and is hard to parallelise, is to break nested functional relationships judiciously by introducing new variables ( the auxiliary coordinates) as equality constraints, and then to optimise a penalised function using alternating optimisation over the original parameters (W step) and over the coordinates (Z step).  The minimisation (W step) updates the parameters by splitting the nested model into independent submodels and training them using existing algorithms, and the coordination (Z step) ensures that corresponding inputs and outputs of submodels eventually match.  In this paper, the basic assumptions of ParMAC are that with large datasets in distributed systems, it is imperative to minimise data movement over the network because of the communication time generally far exceeds the computation time in modern architectures. Thus, the authors propose the ParMAC to translate the parallelism inherent in MAC into a distributed system by data parallelism and model parallelism. They also analyse its parallel speedup and convergence, and demonstrated it with MPI-based implementation to optimise binary autoencoders. The proposed ParMAC is tested on 3 colour image retrieval datasets. 

The organization of the paper is well written, and the presentation is clear. My questions are included in the following:
- The MAC framework solves the original problem approximately. If people use the sigmoid function to smooth the stepwise function, the naive optimization methods can be easier applied. What is the difference between these two? Or why do we want to use a new approach to solve it?
- The authors do not compare their ParMAC model with other distributed approaches for the same nested function optimization problem.

[Official Review · AnonReviewer1 · rating 6 · confidence 4 · 21 Dec 2016]
**Nice idea but not fully fleshed out**

This paper proposes an extension of the MAC method in which subproblems are trained on a distributed cluster arranged in a circular configuration. The basic idea of MAC is to decouple the optimization between parameters and the outputs of sub-pieces of the model (auxiliary coordinates); optimization alternates between updating the coordinates given the parameters and optimizing the parameters given the outputs. In the circular configuration. Because each update is independent, they can be massively parallelized.

This paper would greatly benefit from more concrete examples of the sub-problems and how they decompose. For instance, can this be applied effectively for deep convolutional networks, recurrent models, etc? From a practical perspective, there's not much impact for this paper beyond showing that this particular decoupling scheme works better than others. 

There also seem to be a few ideas worth comparing, at least:
- Circular vs. parameter server configurations
- Decoupled sub-problems vs. parallel SGD

Parallel SGD also has the benefit that it's extremely easy to implement on top of NN toolboxes, so this has to work a lot better to be practically useful. 

Also, it's a bit hard to understand what exactly is being passed around from round to round, and what the trade-offs would be in a deep feed-forward network. Assuming you have one sub-problem for every hidden unit, then it seems like:

1. In the W step, different bits of the NN walk their way around the cluster, taking SGD steps w.r.t. the coordinates stored on each machine. This means passing around the parameter vector for each hidden unit.
2. Then there's a synchronization step to gather the parameters from each submodel, requiring a traversal of the circular structure.
3. Then each machine updates it's coordinates based on the complete model for a slice of the data. This would mean, for a feed-forward network, producing the intermediate activations of each layer for each data point.

So for something comparable to parallel SGD, you could do the following: put a mini-batch of size B on each machine with ParMAC, compared to running such mini-batches in parallel. Completing steps 1-2-3 above would then be roughly equivalent to one synchronized PS type implementation step (distribute model to workers, get P gradients back, update model.)

 It would be really helpful to see how this compares in practice. It's hard for me to understand intuitively why the proposed method is theoretically any better than parallel SGD (except for the issue of non-smooth function optimization); the decoupling also can fundamentally change the problem since you're not doing back-propagation directly anymore, so that seems like it would conflate things as well and it's not necessarily going to just work for other types of architectures.

[Official Review · AnonReviewer4 · rating 4 · confidence 2 · 02 Jan 2017 (modified: 18 Jan 2017)]
**Unclear presentation and some contributions**

UPDATE:
I looked at the arxiv version of the paper. It is much longer and appears more rigorous. Fig 3 there is indeed more insightful.
However, I am reviewing the submission and my overall assessment does not change. This is not a minor incremental contribution, and if you want to compress it into a conference submission of this type, I would recommend choosing message you want to convey, and focus on that. As you say, "...ICLR submission focus on the ParMAC algorithm...", I would focus on this properly - and remove or move to appendix all extensions and theoretical remarks, and have an extra page on explaining the algorithm. Additionally, make sure to clearly explain the relation of the arxiv paper, in particular that the submission was a compressed version.

ORIGINAL REVIEW:
The submission proposes ParMAC, based on MAC (Method of Auxiliary Coordinates), formulating a distributed variant of the idea.

Related Work: In the part on convex ERM and methods, I would recommend citing general communication efficient frameworks, COCOA (Ma et al.) and AIDE (Reddi et al.). I believe these works are most related to the practical objectives authors of this paper set, while number of the papers cited are less relevant.

Section 2, explaining MAC, is quite clearly written, but I do not find part on MAC and EM particularly useful.

Section 3 is much less clearly written. I have trouble following notation, particularly in the speedups part, as different symbols were introduced at different places. Perhaps a quick summary or paragraph on notation in the introduction would be helpful. In paragraph 2, you write as if reader knew how data/anything is distributed, but this was not mentioned yet; it is specified later. It is not clear what is meant by "submodel". Perhaps a more precise example pointing back to eqs (1) & (2) would be useful. As far as I understand from what is written, there are P independent sets of submodels, that traverse the machines in circular fashion. I don't understand how are they initialized (identically?), and more importantly I don't understand what would be a single output of the algorithm (averaging? does not seem to make sense). Since this is not addressed, I suppose I get it wrong, leaving me to guess what was actually meant. 
The fact that I am not able to understand what is actually happening, I see as major issue.

I don't like the later paragraphs on extensions, model for speedup, convergence and topologies. I don't understand whether these are novel contributions or not, as the authors refer to other work for details. If these are novel, the explanation is not sufficient, particularly speedup part, which contains undefined quantities, e.g. T(P) (or I can't find it). If this is not novel, It does not provide enough explanation to understand anything more, compared with a its version compressed to 1/4 of its size and referring to the other work. The statement that we can recover the original convergence guarantees seems strong and I don't see why it should be trivial to show (but author point to other work which I did not look at). In topologies part, claiming that something does "true SGD", without explaining what is "true SGD" seems very strange. Other statements in this section seem also very vague and unjustified/unexplained.

Experimental section seems to suggest that the method is interesting for binary autoencoders, but I don't see how would I conclude anything about any other models. ParMAC is also not compared to alternative methods, only with itself, focusing on scaling properties.

Conclusion contains statements that are too strong or misleading based on what I saw. In particular, "we analysed its parallel speedup and convergence" seems ungrounded. Further, the claim "The convergence properties of MAC remain essentially unaltered in ParMAC" is unsupported, regardless of the meaning of "essentially unchanged".

In summary, the method seems relevant for particular model class, binary autoencoders, but clarity of presentation is insufficient - I wouldn't be able to recreate the algorithm used in experiments - and the paper contains a number of questionable claims.

[Official Review · AnonReviewer3 · rating 5 · confidence 4 · 03 Jan 2017]

The paper presents an architecture to parallelize the optimization of nested functions based on the method of auxiliary coordinates (MAC) (Carreira-Perpinan and Wang, 2012). This method decomposes the optimization into training individual layers and updating the auxiliary coordinates. The paper focuses on binary autoencoders and proposes to partition the data onto several machines allowing the parameters to move between machines. Relatively good speedup factors are reported especially on larger datasets and a theoretical model of performance is presented that matches with the experiments.

My main concern is that even though the method is presented as a general framework for nested functions, experiments focus on a restricted family of models (i.e. binary autoencoders with linear or kernel encoders and linear decoders) with only two components. While the speedup factors are encouraging, it is hard to get a sense of their importance as the binary autoencoder model considered is not well studied by other researchers and is not widely used. I encourage the authors to apply this framework to more generic architectures and problems.

Questions:
1- Does this framework apply to some form of generic multi-layer neural network? If so, some experimental results are useful.
2- What is the implication of applying this framework to more than two components (an encoder and a decoder) and non-linear components?
3- It is desired to see a plot of performance as a function of time for different setups to demonstrate the speedup after convergence. It seems the paper only focuses on the speedup factors per iteration. For example, increasing the mini-batch size may improve the speed per iteration but may hurt the convergence speed.
4- Did you consider a scenario where the dataset is too big that storing the data and auxiliary variables on multiple machines simultaneously is not possible?

The paper cites an ArXiv manuscript with the same title by the authors multiple times. Please make the paper self-contained and include any supplementary material in the appendix.

I believe without applying this framework to a more generic architecture beyond binary autoencoders, this paper does not appeal to a wide audience at ICLR, hence weak reject.

[Author Response · Miguel A. Carreira-Perpinan · 16 Jan 2017]
**RESPONSE TO REVIEWERS**

We thank the reviewers for their reviews. Below we reply individually to each one. Here, we address a comment that several reviewers made, namely that the binary autoencoder model we explore experimentally is not well known by other researchers. It is true that this model is less well known than deep nets, but it was a good choice for this paper for several reasons:
- This type of binary autoencoders is actually well known in the area of binary hashing, where one wants to learn a fast hash function (e.g. linear) with binary outputs because the goal is to do fast image searches in large image databases or similar retrieval problems. We have worked in this area using the MAC algorithm and it was convenient for us to develop ParMAC for it.
- The binary autoencoder allows us to highlight the ability of ParMAC to train non-differentiable models, for which the chain rule does not apply.
- The binary hashing application also provides with large, public training sets (100 million images). This allowed us to test ParMAC in a realistic distributed setting (up to 128 processors over a network). For us it was important to get actual experimental numbers in a distributed cluster (rather than on cores in a machine or simulating network delays).

Finally, perhaps it is not obvious, but implementing and debugging the algorithm in C and MPI costs significant effort, and running the experiments in the UCSD cluster costs real money (around $0.03 per processing core per hour, which quickly becomes hundreds of dollars). This isn't your usual Matlab or GPU experiment... For a team of one student and one faculty member this puts limitations on the size and number of the experiments.

We provide the full C/MPI code in our website to recreate the experiments in either a shared- or a distributed-memory system.

[Final Decision · Program Chairs · 06 Feb 2017]
**ICLR committee final decision**

The work proposes a parallel/distributed variant of the MAC decomposition method. In presents some theoretical and experimental results supporting the parallelization strategy. The reviews are mixed and indeed a common concern among the reviewers was the choice of test problem. To me it is ok to only concentrate on a single class of problems, but in this case it needs to be a problem that the ICLR community identifies as being of central importance. Otherwise, if a more esoteric problem is chosen then I (and the reviewers) would rather see that the method is useful on multiple problems. Otherwise, it's basically impossible to extrapolate the experiments to new settings and we are forced to re-implement the algorithm. I'm not saying that the authors necessarily need to consider deep networks and there are many alternative possible models (sparse coding, collaborative filtering, etc.). But it should be noted that, without further experimental comparisons, it is impossible to verify the author's claims that the method is effective for deeply-nested models.
 
 Other concerns brought up by the reviewers (beyond the clarity/presentation issues, which should also be addressed): the experimental comparison would be more convincing with a comparison to an existing approach like a parallel SGD method. I appreciate that the authors have done a lot of work already on this problem, but doing such obvious comparisons should be the job of the author instead of the reader (focusing purely on parallelization would be ok if the MAC model was extremely-widely-used already and parallelizing was an open problem, but my impression is that this is not the case). As a minor aside, the memory issue will be more serious for deeply-nested models, due to the use of the decomposition approach (we don't want to store the activations for all layers for all examples), and this doesn't arise in SGD.